# Measuring DNI with a New Radiometer Based on an Optical Fiber and Photodiode

**DOI:** 10.3390/s24113674

**Published:** 2024-06-06

**Authors:** Alejandro Carballar, Roberto Rodríguez-Garrido, Manuel Jerez, Jonathan Vera, Joaquín Granado

**Affiliations:** 1Electronic Engineering Department, Universidad de Sevilla (E.T.S. de Ingeniería), C/Camino de los Descubrimientos s/n, 41092 Sevilla, Spain; mjerez@us.es (M.J.); j_granado@us.es (J.G.); 2Capgemini Engineering, Edificio Oxxeo, C/Puerto de Somport 9, 28050 Madrid, Spain; roberto.rodriguez.garrido@capgemini.com; 3National Renewable Energy Center (CENER), Solar Thermal & Thermal Energy Storage Department, C/Isaac Newton, 4. Pabellón de Italia, Planta 5 SO, 41092 Sevilla, Spain; jvera@cener.com

**Keywords:** radiometer, pyrheliometer, solar irradiance, optical fiber, semiconductor photodiode, spectral irradiance, acceptance cone

## Abstract

A new cost-effective radiometer has been designed, built, and tested to measure direct normal solar irradiance (DNI). The proposed instrument for solar irradiance measurement is based on an optical fiber as the light beam collector, a semiconductor photodiode to measure the optical power, and a calibration algorithm to convert the optical power into solar irradiance. The proposed radiometer offers the advantage of separating the measurement point, where the optical fiber collects the solar irradiation, from the place where the optical power is measured. A calibration factor is mandatory because the semiconductor photodiode is only spectrally responsive to a limited part of the spectral irradiance. Experimental tests have been conducted under different conditions to evaluate the performance of the proposed device. The measurements confirm that the proposed instrument performs similarly to the expensive high-accuracy pyrheliometer used as a reference.

## 1. Introduction

Solar energy stands out as an inexhaustible, clean, and renewable energy resource in the world, which is key in the energy transition toward sustainability. Since the establishment of the foundation for near-zero-emission buildings by the European Union directive [1,2], data related to solar irradiation have become critical for making informed investments decisions [3,4]. Additionally, other applications, such as agriculture; energy balance within climate change and oceanography; health implications, including the effects of ultraviolet radiation on the skin; and endeavors for renewable energy for sustainability [5], also necessitate accurate solar irradiation information [6].

The measurement of radiant flux received by a surface per unit area (using the SI unit W/m^2^) is known as irradiance, while solar radiation is the energy of the sun that falls on Earth's surface. Spectral irradiance ranges from 280 nm to 4000 nm, with between 97% and 99% of its spectral components concentrated between 300 nm and 3000 nm [7,8]. Spectral irradiance changes over time during the day, predominantly at dawn and dusk, as well as with location, and comes from different parts of the sky [9].

Irradiance can be decomposed into two components, i.e., direct beam (known as direct normal irradiance—DNI) and diffuse beam (known as diffuse horizontal irradiance—DHI). The sum of both is called the global horizontal irradiance (GHI) [7,8]. Devices measuring solar radiation are named radiometers [10,11,12,13,14,15], classified according to which beam is measured. Pyrheliometers are used to measure the DNI, consisting of a radiometer mounted on a sun tracker [16,17], while pyranometers are mainly used for measuring the GHI and DHI [18,19].

The key element of these devices is their sensor, with passive thermoelectric or photoelectric sensors predominantly used to convert irradiance into an equivalent difference of voltage. Currently, there is only one type of commercially available pyrheliometer, based on a thermopile, that is used to convert thermal energy into a voltage difference. The main advantage of this type of pyrheliometer is that it measures with a flat spectral sensitivity between 300 and 3000 nm. However, its main drawback is the high cost associated with such a device [6].

On the other hand, semiconductor detectors have a big problem with the great affections caused by temperature, so many authors have been forced to use temperature stabilization systems to mitigate the problem and conveniently measure the GHI [20]. Other options, such as the use of an optical fiber to pass sunlight to a photodiode for measuring the GHI, have been proposed [21,22,23].

This work proposes a fast-response and cost-effective radiometer for measuring the DNI. With this instrument, the solar irradiation is collected at one end of a multimode fiber-optic cable, guided along the fiber, and converted into an electric current in a semiconductor photodiode. A complementary correction factor [9,24] for converting the measured optical power into DNI is proposed. The response of the proposed sensor was experimentally evaluated and compared to that of a thermopile-based pyrheliometer mounted on the same sun tracker. The experimental results confirm that the proposed sensor performs similarly to the reference.

The remainder of this paper is organized into four sections. Section 2 describes the block diagram and the principle of operation of the proposed instrument. Section 3 presents the necessary correction factor to convert optical power into solar irradiation. Section 4 outlines the experimental setup along with the measurement results, discussing its advantages and drawbacks. Finally, some conclusions are drawn in Section 5.

## 2. System Description

Solar light is completely described by its spectral irradiance E(λ), which changes depending on the geographical location of the measurement, as well as on the time (season, month, day, and hour) [9,25,26]. The total irradiance, or simply irradiance, is the integral of spectral irradiance along the whole spectral measuring range.

ISO-9488 [7] defines direct solar irradiance as the “*quotient of the radiant flux on a given plane receiver surface from a small solid angle centered on the sun’s disk to the area of that surface. If the plane is perpendicular to the axis of the solid angle, direct normal solar irradiance is received*”. Following this definition, the principle of operation of the proposed radiometer lies in the estimation of the DNI by using the relationship between the solar optical power captured by the core of a multimode optical fiber and its surface area from the solid angle defined by the fiber numerical aperture. However, as stated in [6], the DNI definition in [7] remains vague due to the lack of a specification about the magnitude of the “small solid angle”. The aperture of the solid angle recommended to measure the DNI is not standardized and continues to be a topic of discussion, ranging from half-apertures from 2.5° to 10° [6]. Let us denote *B_n_* as the experimental DNI.

The block diagram of the proposed device is depicted in Figure 1. The solar light is collected by an open-ended multimode fiber-optic cable and guided to an optical power meter (OPM) where the measurement of the coupled optical power is performed and registered by an acquisition module (DAQ).

The geometry and dimensions of the fiber-optic end (which is exposed to the solar radiation) are detailed in Figure 2, where *a* and *b* are the radius of the core and the cladding, respectively. Furthermore, the refractive indexes, *n*_1_ for the core and *n*_2_ for the cladding, determine the fiber-optic numerical aperture (NA), where *θ*_a_ represents the half-acceptance angle of the optical fiber, as shown in Figure 2:(1)$$NA=sen\left(\theta_{a}\right)=\sqrt{n_{1}^{2}-n_{2}^{2}}$$

The proposed device collects the solar beam inside the fiber-optic acceptance cone, delimited by *θ*_a_, and travels the fiber-optic length, *L*, until reaching the optical power meter. To prevent potential measurements degradation caused by temperature exposure, the semiconductor photodiode is located far from the collection point. Thus, a non-negligible propagation loss of *α·L* must be considered, where *α* (*λ*) is the wavelength-dependent fiber-optic attenuation coefficient.

The optical power meter is based on a semiconductor photodiode (Sc-PD) sensor. The Sc-PD supplies the so-called photocurrent, which is proportional to the coupled optical radiation falling on the fiber-optic core area. The conversion from the incident optical power into photocurrent is determined by the Sc-PD responsivity, R, which is wavelength-dependent. Selecting a specific responsivity value for a reference wavelength *λ_i_*, R (*λ_i_*), this photocurrent is used by the OPM to provide an estimation of the optical power measurement, *P* (*λ*_i_), given by [27]:(2)$$P\left(\lambda_{i}\right)=\frac{1}{{R(\lambda }_{i})}\int_{\lambda_{min}}^{\lambda_{max}}{E\left(\lambda \right)\cdot e}^{-\alpha \left(\lambda \right)\cdot L}\cdot R\left(\lambda \right)\cdot d\lambda$$
where *λ_min_* and *λ_max_* define the minimum and maximum wavelengths for the solar spectral range, respectively. It is important to note that, although the photodetector defines a responsivity curve between two wavelengths, the OPM only uses one specific value of the responsivity to perform the conversion from the electric signal to the value of the optical power measurement, *P* (*λ*_i_). This means that the power detected by the OPM depends on the selected wavelength in the OPM, *λ_i_*, which needs to be a value within the operating range of the photodiode. During the calibration process, this will be corrected to consider all the photodiode detecting range.

Finally, the data acquisition module of Figure 1 registers and stores the optical power traces provided by the OPM with a configurable sampling rate. Once the data are stored, an estimation for the DNI is computed by the calibration algorithm described in the following section.

## 3. Calibration Process

The calibration process is illustrated in Figure 3. As explained below, this procedure is mandatory to provide an estimation for the DNI and can be implemented either in a dedicated microprocessor connected directly to the OPM, enabling real-time DNI measurements, or in a computer for off-line processing.

Once the measurement of the optical power trace *P* (*λ*_i_) is performed by the OPM, a preliminary estimation of the solar irradiance *Φ_raw_ (λ_i_)* is conducted, as obtained in (3):(3)$$\Phi_{raw}\left(\;\lambda_{i}\right)=\frac{P(\lambda_{i})}{A_{eff}}$$
where *A*_eff_ represents the fiber-optic effective area, i.e., only a portion of the incident solar beams are properly coupled to the fiber and converted into current in the photodiode. For a multimode fiber, this area can be specified by the core area, *A_eff_* = *π*·*a*^2^, as shown in Figure 2.

However, this raw estimation of solar irradiance is far from the DNI measurement. On one hand, this raw estimation only accounts for a spectral portion of the solar radiation due to the limited spectral SC-PD response compared to the complete spectral solar irradiance. On the other hand, both the photodiode responsivity and the solar spectral irradiance strongly depends on the wavelength, as shown in Figure 4. Therefore, it is mandatory to compensate for the raw irradiance before determining the DNI measurement.

We propose the correction factor *CF* (*λ*_i_) computed by (4) to compensate for the effects of wavelength dependence of the SC-PD response, the solar spectral irradiance, and the insertion loss of the optical fiber [27]:(4)$$CF(\lambda_{i})=\frac{\int_{\lambda_{min}}^{\lambda_{max}}E_{ref}\left(\lambda \right)\;d\lambda }{\int_{\lambda_{min}}^{\lambda_{max}}E_{ref}\left(\lambda \right)\;e^{-\alpha \left(\lambda \right)L}R\left(\lambda \right)\;d\lambda }R\left(\lambda_{i}\right)$$
where:*E_ref_* (*λ*) is a spectral irradiance pattern of the solar beam (taken from ASTM G173-03 Reference Spectra derived from SMARTS [28]);$R\left(\lambda \right)$ is the responsivity of the semiconductor photodiode;*λ*_i_ is the wavelength selected in the OPM to convert the current into optical power;α (λ) is the attenuation coefficient per length unit of the optical fiber;*L* is the length of the fiber-optic cable from the tip exposed to the solar beam to the photodiode detector.

Finally, a measurement for the DNI, *B_n_*, can be calculated by multiplying the raw solar irradiance in (3) by the computed correction factor obtained in Equation (4):(5)$$B_{n}=\Phi_{raw}\left(\;\lambda_{i}\right)\cdot CF\left(\;\lambda_{i}\right)$$

## 4. Experimental Measurements

### 4.1. Setup Description

A proof of concept of the proposed sensor was conducted at the facilities of the University of Seville. The testing facilities were provided by the Group of Thermodynamics and Renewable Energy (GTER) of Universidad de Sevilla. This group possesses a weather station, located at the coordinates 37.41 N, 6.01 W at 12 m of elevation, and it is composed of different radiometric instruments for DNI, DHI and GHI measurements. The complete information about their facilities and all their instruments, as well as real-time measurements and forecasts from this station, can be found at their website (http://estacionmeteo.us.gter.es/, accessed on 15 April 2024).

Four different optical fibers as capturing elements for the solar beam were used, whose details are included in Table 1. They were a multimode fiber of 50/125 μm with 0.22 NA (model FG050LGA), two multimode fibers of 105/125 μm with 0.1 NA and 0.22 NA (models FG105LVA and FG105LCA, respectively), and a 200/225 μm multimode fiber with 0.50 NA (model FP200URT). The fiber cables were provided by Thorlabs (Newton, NJ, USA).

The testbench is shown in Figure 5. Figure 5a shows the end of the fiber optic cable that was exposed to the solar radiation. It was positioned on the building roof and mounted on a sun tracker (model Solys Gear Drive) using feedthroughs directly placed on the pyrheliometer structure to ensure alignment with the solar disk. Three pyrheliometers were used as a reference for the DNI measurement. Among them, the results displayed and discussed throughout this paper correspond to the comparison with the pyrheliometer CHP1, which was installed on the same sun tracker as the tip of the fiber. This CHP1 pyrheliometer provided the most accurate DNI measurements. When compared to the proposed capturing element, this pyrheliometer has a more reduced field of view (an opening half-angle of 2.5°, much narrower than the fiber-optic half-acceptance angles used in the proposed radiometer) and a slow response time, which goes up to 5 s, due to the use of a thermopile as a detecting element instead of a semiconductor photodiode. The fact that the field of view of the proposed instrument is wider than that of the CHP1 pyrheliometer makes the device more robust to potential alignment errors of the solar tracker, ensuring that even with minor mechanical faults, the solar disk remains within the field of view of the proposed device. The sun tracker and pyrheliometer were provided by Kipp & Zonen (Sterling, VA, USA).

The opposite end of the optical fiber, which were 10 m long, were connected to a silicon photodiode (model S140C) located inside the laboratory, as shown in Figure 5b. Whereas the tip of the optical fiber exposed to radiation was placed in the solar tracker, the photodiode, the OPM and the computer were placed in an indoor laboratory on the floor below the weather station. The responsivity of the photodiode, ranging from 350 nm to 1100 nm, was provided by the manufacturer, which is shown in Figure 4. The photodiode was connected to an optical power meter (PM320E). The photodiodes and the power measurement instrument were provided by Thorlabs.

### 4.2. Results

This section presents a comparative analysis between the direct normal irradiance measurements obtained from various configurations of the proposed radiometer and those derived from the commercial pyrheliometer. Besides the different configurations for the device, three types of weather conditions are analyzed in the discussion of the results, which are sunny, cloudy, and rainy days, with the purpose of discussing the influence of weather changes on the accuracy of the measurements [29]. The need for accurate measurements in those three general conditions is associated with the possible applications of the proposed radiometer. When using the radiometer in CSP plants, measuring the DNI with a clear sky is crucial for monitoring the position of the heliostats and enhancing the strategies to optimize energy generation. On the other hand, monitoring the DNI is important in photovoltaic applications, where the electricity generated is influenced by the different meteorological conditions.

Figure 6 shows the DNI measurements obtained from both the commercial pyrheliometer and the proposed instrument configured with two different multimode fibers during two mostly sunny days: FG050LGA and FP200URT. For the optical fiber FG050LGA, presented in Figure 6a, a working wavelength of *λ_i_* = 635 nm was selected in the OPM to measure the coupled optical power, resulting in a correction factor of 1.30, whereas, in Figure 6b, a second optical fiber, FP200URT, is evaluated, with *λ_i_* = 865 nm selected in the OPM, resulting in a correction factor of 2.03. The experimental results demonstrate that the proposed device performs comparably to the commercial pyrheliometer used as a reference.

The measurements shown in Figure 6 also show different ripples depending on the fiber core diameter. The results obtained with the fiber FP200URT (core diameter of 200 μm) exhibit minor ripples when compared to those obtained with the fiber FG050LGA (core diameter of 50 μm). The larger the core diameter is, the smaller the ripples are in the DNI provided by the new instrument since the coupled optical power in the optical fiber increases. The measurements shown in Figure 6b include some periods of time where clouds are present, while in Figure 6a, the sky was clear for the entire day. In such a situation, the proposed device performs similarly to the commercial one. However, the differences between the DNI provided by the commercial pyrheliometer and the new instrument lies in the correction factor, which presents a direct dependence on the spectral irradiance pattern taken as a reference [9,25,26,29].

For the results presented in Figure 7a, a multimode fiber FG105LVA was used during a completely sunny day. In this case, the fiber used as the capturing element had a numerical aperture of 0.1, which translates into a half-angle aperture of less than 6°, which adjusts to some of the definitions for pyrheliometer given by [6], and the results endorse its good performance. In terms of ripple, the proposed instrument performed similarly to the commercial pyrheliometer. Figure 7b represents the absolute and relative deviations obtained from the DNI measurement shown in Figure 7a. The experimental results demonstrate that the proposed device exhibited higher accuracy during the central hours of the day, keeping a relative deviation below 5% between 10:00 and 19:00 h. However, the comparison between the DNI provided by the commercial pyrheliometer and the new instrument exhibits a small divergence during sunrise and sunset. The main error occurred around time intervals centered at 9:00 and 20:00 h, which correspond to sunrise and sunset. These periods of time are precisely when the solar radiation passes through a higher portion of air mass, thus making the solar spectral irradiance more sensitive. As it has been said, the solar spectrum model used to compute the correction factor, *E_ref_ (λ)*, is constant and independent of the time of day and weather, i.e., the ASTM G173-03 Reference Spectra [28], although, in fact, it should be subject to variations due to these factors. In this sense, it is important to note that the introduction of a correction factor depending on the solar spectral irradiance pattern throughout the day will improve the accuracy of the DNI measurement. 

Throughout all the results presented, the correction factor varied among different values as a function of the selected wavelength in the OPM, which resulted in some corrections amplifying the raw measurements and others attenuating it. As presented in Equation (4), the correction factor depends only on the solar spectrum model and the responsivity at the reference wavelength of the photodiode, as the attenuation is negligible for small distances. As the spectral irradiance model, *E_ref_* (*λ*) was the same in all the conditions, the variation in the correction factor depends only on the responsivity of the photodiode at *λ*_i_, meaning that the correction factor describes a curve dependent on the reference wavelength similar to the responsivity, as Figure 8 represents. From the analysis of this curve, it is visible that two values exist that make the correction factor equal to 1, which, if used in the OPM, would make the raw irradiance value equal to the calibrated one, easing the computation. In the case of the silicon photodiode S140C from Thorlabs, these values are 564 nm and 1054 nm.

To assess the accuracy of the measurements in other meteorological conditions and to evaluate the response time and robustness of the proposed radiometer, some results from cloudy and rainy days have been evaluated. Figure 9 shows the measured DNI in a day with the intermittent presence of clouds. For this setup, two multimode fibers, FG050LGA and FG105LCA, were used. For the optical fiber FG050LGA, shown in Figure 9a, a working wavelength of *λ*_i_ = 635 nm was selected in the OPM to measure the coupled optical power (correction factor of 1.30 as in Figure 6a), whereas in Figure 9b, the optical fiber FP200URT was evaluated with *λ*_i_ = 400 nm selected in the OPM, resulting in a correction factor of 0.26. The influence of clouds significantly affected the measurements, resulting in fluctuations in the DNI trace throughout the day. Note that, despite the higher ripple associated with its smaller core, the results obtained by the FG050LGA fiber (see Figure 9a) outperformed those obtained with the FG105LCA.

The DNI measurement results for another cloudy day using the Thorlabs FG105LVA optical fiber are presented in Figure 10a, as are the absolute (Figure 10b) and relative (Figure 10c) deviations between the DNI measurements provided by the proposed radiometer and the commercial pyrheliometer. The results are similar to those observed previously, where the error was minimized during the central hours of the day and grew at the start and end of the day. In this case, the deviation during the passage of clouds was higher than when the sky was clear, as the response to the shadowing and clearing of the sky was not exactly the same and due to small variations in the spectral irradiance. Having said that, the relative deviations were kept below 7% most of the time between 9:00 and 19:00, proving great accuracy also in cloudy conditions.

Before the final discussion, it is worth noting the operating conditions and behavior of the proposed device against the commercial pyrheliometer during rain. In Figure 11, the DNI measurements during a 10 min interval under rainy conditions are presented. The measurements correspond to a particular day after a week of continuous measurements in rainy and hazy conditions. Figure 11 depicts how rapid changes in the irradiance are not visible for the commercial pyrheliometer, while they are for the proposed radiometer. This is due to the difference between the detecting elements in both instruments, where the silicon photodiode has a much quicker response time, in the order of microseconds, compared to the 5 s limit by the commercial device.

These results also indicate the reliability, long-term stability, and robustness of the proposed instrument. In this sense, both the commercial pyrheliometer and the proposed radiometer are sensible to the deposition of dust, water, and other substances on the exposed surface to the sun, overshadowing the field of view of the sensors. As this surface is much smaller in the case of the fiber compared to the commercial pyrheliometer, the likeliness of this happening is very improbable and can be prevented with regular cleaning of the tip of the fiber, which might need to be carried out more regularly with the pyrheliometer window.

### 4.3. Discussion

The experimental results described in the previous section validate the proposed instrument performance across various types of fiber, several working wavelengths, and different meteorological conditions. Upon comparing different types of fiber, we can conclude that the fiber with a narrower core, i.e., 50/125, yielded measurements closest to those obtained by the reference pyrheliometer, despite exhibiting greater ripples in the DNI measurement. If we consider the numerical aperture, the most accurate to the results given by the pyrheliometer were those with AN 0.1, as the half-angle acceptance of 5.7° is comparable with the 2.5° used by the commercial device.

The results presented demonstrate that raw solar irradiance measurements may deviate significantly from those provided by the commercial pyrheliometer used as a reference. Thus, the so-called correction factor plays a key role in correcting this deviation. In this paper, the solar spectral irradiance pattern used for calculating the correction factor, E_ref_ (λ), was assumed to be constant, equivalent to the value at midday under clear sky conditions given by the ASTM G173-03 Reference Spectra derived from SMARTS [28]. However, the solar irradiation spectrum is affected by fluctuations. Although the spectrum varies due to atmospheric conditions, it has been proven that the measurements during central hours of the day are accurate even if the conditions are not sunny, as the spectrum does not vary as much as during dawn and dusk, where the main divergencies are located. To enhance the DNI estimation accuracy of the proposed sensor, future work should incorporate a correction factor that would consider the variation in the solar spectral irradiance throughout the day and the atmospheric conditions.

Related to the instrumental error, two main sources of error must be considered. On one hand, there is a random error associated with the optical power measurement uncertainty provided by the photodiode datasheet, which is 3.6% of the solar light we are addressing. This error, however, is minimized, since each value in the power trace generated by the OPM for the results presented corresponds to the averaged value of 10 samples per second, reducing this uncertainty to approximately 1%.

The other main source of error is the tolerance in the manufacturing of the optical fiber, with core diameters that can vary from the theoretical value between 2% and 3% depending on the model. This error has a greater influence on the final DNI value, but as it is a systematic and constant error, it can be eliminated by calibrating the instrument against a standard reference. This strategy has not been implemented in the presented results.

Given this information and using the fundamentals of the theory of measurement errors [30], the maximum instrumental error is 8.6% in the worst-case scenario. This scenario involves using the fiber LG105LVA, which has the greatest diameter variability, and considering 3.6% as the error associated with the photodiode (assuming the samples are not averaged when generating the power trace). This value was calculated using a reference maximum DNI value of 900 W/m^2^ and a correction factor of 1. This maximum value is greater to the CHP1 measurement error, which is 3% per hour and 2% per day, but it could be minimized by introducing the power averaging and correction factor mentioned above, reaching comparable values.

The experimental results demonstrate that the proposed instrument exhibited a fast response, which is comparable to the commercial pyrheliometer, enabling it to track typical temporal variations in the solar irradiation due to presence of clouds. In addition, the proposed instrument offers advantages in terms of its cost and dynamic range. 

Concerning its costs, the proposed radiometer has a great competitive advantage when used as a pyrheliometer, since its cost is approximately one-third of that of the commercial pyrheliometer. Relative to the dynamic range, one of the mains strengths of the proposed instrument is its modularity, which allows for selecting different core diameters or numerical apertures for the fiber, as well as choosing different semiconductor photodetectors. These factors allow the proposed instrument to work in different spectrum ranges depending on the desired applications and varying the dynamic range according to the specifications.

## 5. Conclusions

This paper proposes a simple instrument to measure the direct normal solar irradiance by using an optical fiber as the light beam collector and a silicon photodiode to measure the coupled optical power through the fiber core surface. For its proof of concept, the proposed instrument was implemented and compared to a high-performance thermopile-based pyrheliometer, which is significantly more expensive. The experimental DNI measurements demonstrate that both instruments exhibited similar accuracy under different conditions.

The proposed instrument is particularly well-suited for different configurations since the fiber geometry and photodiode sensitivity can be finely tuned to achieve the desired dynamic range. Moreover, its modular design and the decoupling of the capturing and detecting elements are perfectly suited for its use in PV and CSP plants, allowing for installation directly on photovoltaic panels and heliostats and isolating the measurement from extreme environmental conditions, without dismissing its possible use in other fields where radiometers are needed, such as in agriculture or weather stations. 

The imbalance between the photodiode responsivity and the fluctuating solar irradiance challenges are in accurately converting the optical power to solar irradiance. The main enhancements that should be considered are related to the errors and deviations produced by different factors. First, related to the variation in the solar spectrum throughout the day, future work should take into account the dependance of these spectral irradiance references when computing the correction factor, with the aim of solving the deviation between the measurements from the pyrheliometer and the proposed radiometer. Secondly, although they have a smaller impact on the results, errors associated with the instrumentation may be considered as well. Among them, the most important is the relative error in the geometrical features of the optical fiber, with a tolerance between 2% and 3% in the core diameter depending on the model.

At the expense of a slight absolute error when measuring the DNI, the proposed radiometer is characterized by a cost-effective solution, a very fast response, huge versatility and flexibility to choose the acceptance cone to collect solar radiation, a large dynamic range due to the possibility of working with different optical fiber core sizes, optimal sensitivity thanks to its use of a semiconductor photodiode, and high immunity to noise and temperature.

## Figures and Tables

**Figure 1 sensors-24-03674-f001:**
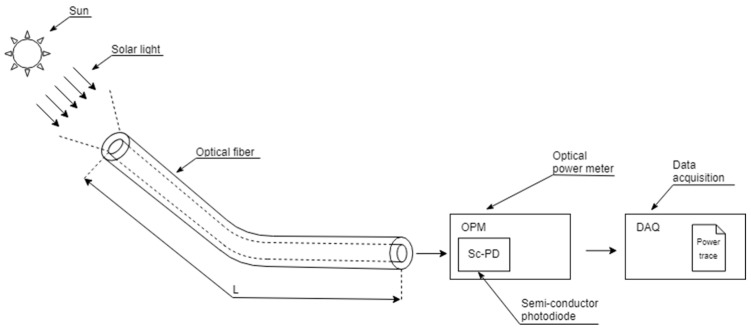
Block diagram for the proposed radiometer. The system is composed of an optical fiber (OF), a semiconductor photodiode (SC-PD), an optical power meter (OPM), and a module for data acquisition (DAQ).

**Figure 2 sensors-24-03674-f002:**
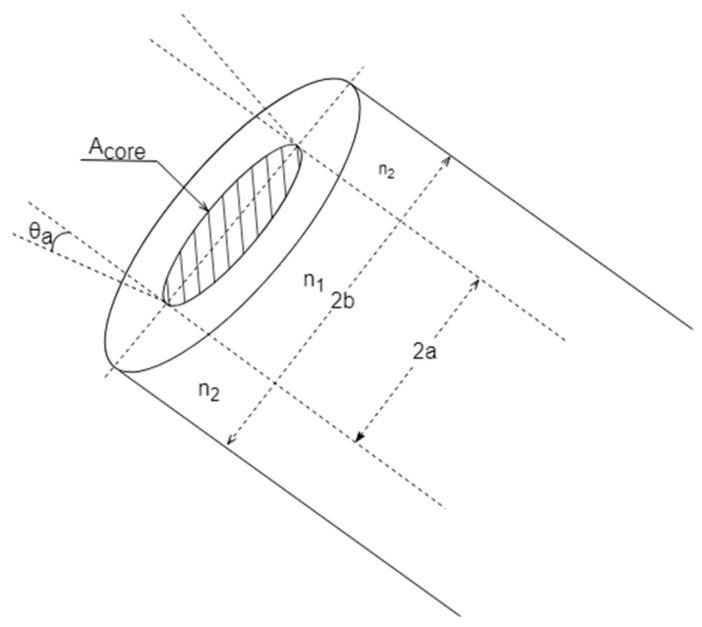
Scheme of the multimode optical fiber’s end tip exposed to the solar radiation.

**Figure 3 sensors-24-03674-f003:**
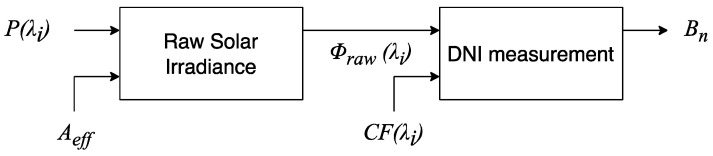
Calibration process: *P* (λ_i_) is the estimated optical power measurement, *A*_eff_ is the fiber-optic effective area, *Φ_raw_* (λ_i_) is the raw solar irradiance, *CF* (*λ*_i_) is the correction factor, and *B_n_* is the resulting DNI measurement.

**Figure 4 sensors-24-03674-f004:**
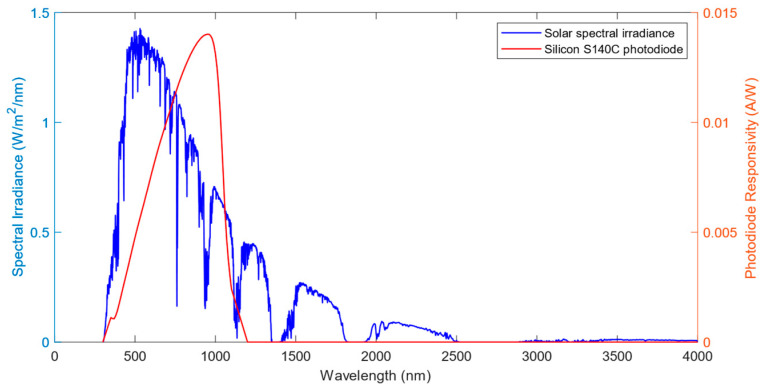
Solar spectral irradiance [28] (blue line), and responsivity response for the silicon photodiode Thorlabs S140C (red line) as a function of wavelength.

**Figure 5 sensors-24-03674-f005:**
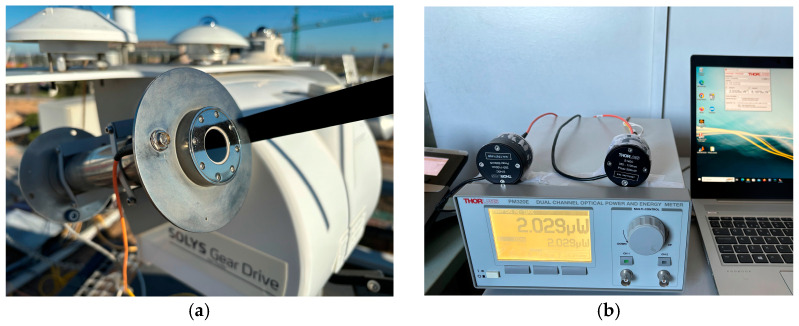
Experimental assembly for the proof of concept of the proposed instrument: (**a**) optical fiber and commercial pyrheliometer installed in a sun tracker; and (**b**) silicon photodiodes and an optical power meter, plus a computer for data acquisition.

**Figure 6 sensors-24-03674-f006:**
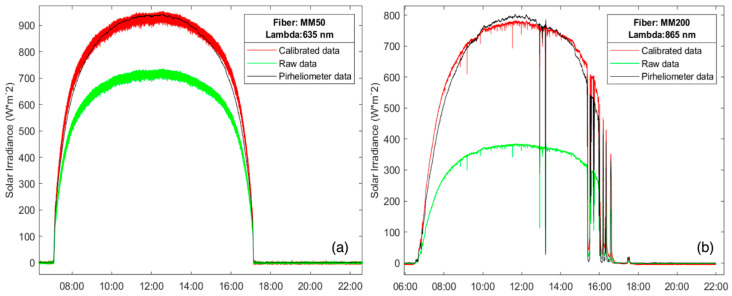
DNI measurements obtained by the proposed radiometer (red line) and the commercial pyrheliometer (black line) for two sunny days, with: (**a**) Thorlabs FG050LGA multimode fiber; and (**b**) Thorlabs FP200URT multimode fiber. The green line represents the raw solar irradiance before applying the correction factor.

**Figure 7 sensors-24-03674-f007:**
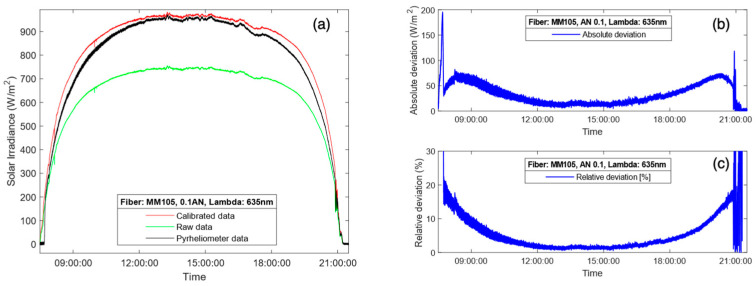
DNI measurements obtained by the commercial pyrheliometer (black line) and the proposed radiometer (red line) with a Thorlabs FG105LVA multimode fiber: (**a**) DNI measurements for a sunny day; absolute (**b**) and relative (**c**) deviations between the DNI measurements provided by the proposed instrument and the commercial pyrheliometer. Green line in (**a**) represents the raw solar irradiance before applying the correction factor.

**Figure 8 sensors-24-03674-f008:**
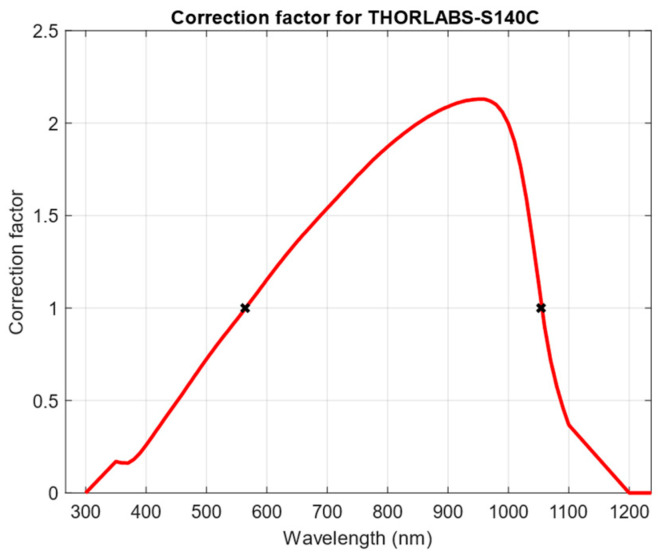
Variation of the correction factor depending on the selected reference wavelength in the OPM.

**Figure 9 sensors-24-03674-f009:**
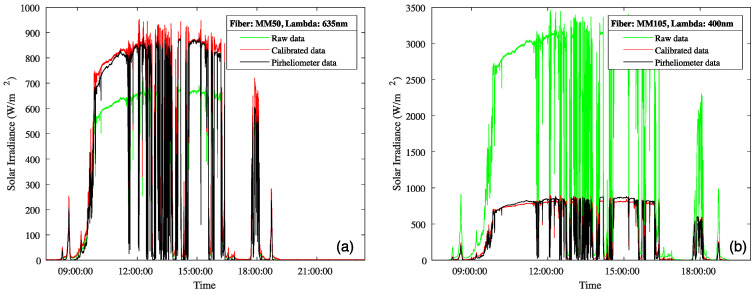
DNI measurements obtained by the commercial pyrheliometer (black line) and the proposed radiometer (red line) on a cloudy day, with: (**a**) Thorlabs FG050LGA multimode fiber; and (**b**) Thorlabs FG105LCA multimode fiber. The green line represents the raw solar irradiance before applying the correction factor.

**Figure 10 sensors-24-03674-f010:**
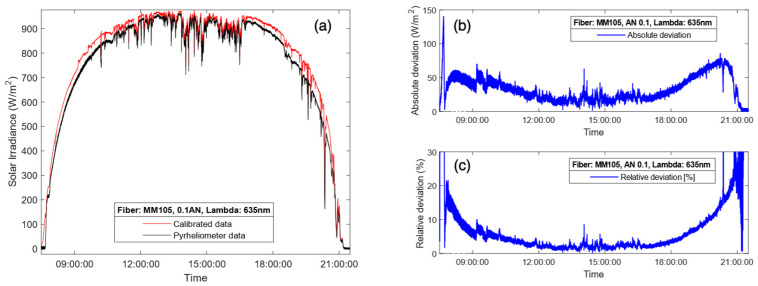
DNI measurements obtained by the commercial pyrheliometer (black line) and the proposed radiometer (red line) with the Thorlabs FG105LVA multimode fiber: (**a**) DNI measurements for a cloudy day; absolute (**b**) and relative (**c**) deviations between the DNI measurements provided by the proposed instrument and the commercial pyrheliometer.

**Figure 11 sensors-24-03674-f011:**
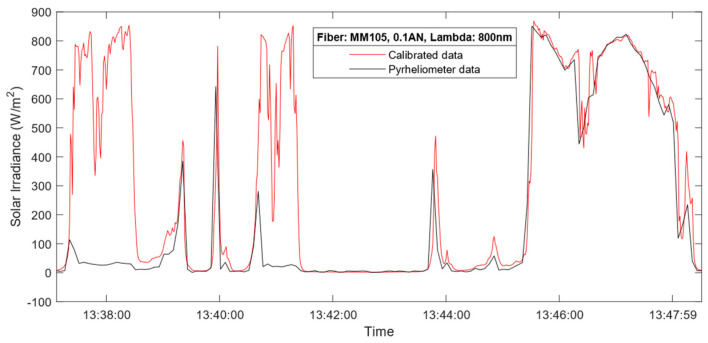
DNI measurements obtained by the commercial pyrheliometer (black line) and the proposed radiometer (red line) for a rainy interval with the Thorlabs FG105LCA multimode fiber.

**Table 1 sensors-24-03674-t001:** Details of the optical fibers used for the proof of concept of the proposed radiometer.

Fiber Model	Core Diameter (µm)	Numerical Aperture	Half-Acceptance Angle
Thorlabs FG050LGA	50	0.22	12.71°
Thorlabs FG105LCA	105	0.22	12.71°
Thorlabs FG105LVA	105	0.1	5.73°
Thorlabs FP200URT	200	0.5	30°

## Data Availability

The raw data supporting the conclusions of this article will be made available by the authors on request.

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
