# Peer review of "Measuring DNI with a New Radiometer Based on an Optical Fiber and Photodiode"

_sensors, 2024, doi:10.3390/s24113674_

Round 1
Reviewer 1 Report
Comments and Suggestions for Authors
The research presents a novel radiometer designed for cost-effective measurement of direct normal solar irradiance (DNI). Utilizing an optical fiber to collect solar irradiation and a semiconductor photodiode for optical power measurement, the device separates the collection and measurement points, enhancing flexibility and practicality. A specific calibration algorithm is employed to convert optical power into solar irradiance, addressing the spectral response limitations of the semiconductor photodiode. The device's performance was tested under various conditions, showing comparable results to those of a high-accuracy, more expensive pyrheliometer.
Comments to the authors for manuscript improvement before publishing:
Enhance Methodological Detail: Provide more detailed descriptions of the experimental setup and conditions under which the device was tested, ensuring reproducibility of the results.
Broader Calibration Range: Consider expanding the calibration to cover a wider range of spectral irradiance to improve the device’s applicability under different solar conditions.
Long-Term Stability Testing: Include data on the long-term stability and reliability of the device to assess its performance over extended periods of use.
Comparative Analysis: Enhance the comparative analysis by including more reference instruments and a broader range of environmental conditions to validate the device's performance more comprehensively.
Error Analysis: Conduct a thorough error analysis to quantify the potential sources of uncertainty in the measurements and how they affect the overall accuracy of the device.
Cost Analysis: Provide a detailed cost analysis comparing the new radiometer to traditional devices, highlighting the cost-effectiveness in terms of both construction and operation.
Practical Deployment Scenarios: Discuss potential practical applications and deployment scenarios for the radiometer to illustrate its utility in real-world solar energy projects.
Future Enhancements: Suggest possible improvements or future enhancements that could be made to the radiometer’s design or functionality to extend its applicability or improve its performance.
Author Response
Dear Reviewer #1,
We appreciate your comments and revisions that helped us to improve the manuscript and our knowledge about the studied phenomenon. The detailed answers to your new questions and comments are shown in the following Annex.
Kind regards.
The authors

Reviewer 2 Report
Comments and Suggestions for Authors
Dear Authors,
congratulations on the article. However, I consider that a number of changes/modifications should be incorporated for its final publication.
1. The acceptance angle of the pyrheliometer and the optical fibers used in the experiments must be similar in order to be able to compare the measurement between them, since if this is not the case, the sensors would not be receiving the same amount of irradiance. In addition to equation 2, where it is specified how to calculate the half-acceptance angle, authors should include the specific value of the half-acceptance angle by both the pyrheliometer and the different optical fibers used in the experiments.
2. The article misses a correct treatment of the errors and uncertainties associated with the measurement, especially when it comes to the development of sensors. What is the instrumental error of the pyrheliometer, the photodiodes and the optical power meter? How is this instrumental error translated into the uncertainty of the final measurements?
A correct treatment of errors should be included in the article.
3. In the caption of Figures 7b and 7c, measurement error is given when, in fact, it is the deviation of the fiber measurements from the pyrheliometer measurement. Please correct it in the caption sentence.
4. As the authors pointed out, the calibration factor of the sensor is deeply dependent on the spectrum of the solar radiation incident on the optical fibers, since the response of the photodiode has a spectral dependence, as shown in Figure 4. Therefore, in the conclusions of the article, a paragraph should be added reflecting the validity of the instrument in its current state: valid only for sunny days around solar noon or, if necessary, derive a calibration factor that is dependent on the solar spectrum and extends its validity to all measurement times, since the deviations with respect to the pyrheliometer measurements shown in Figure 7b and 7c exceed 10% at certain times of the day.
5. The authors should also include a paragraph explaining what happens if the changes in the solar spectrum fall outside the spectral response of the photodiodes.
Author Response
Dear Reviewer #2,
We appreciate your comments and revisions that helped us to improve the manuscript and our knowledge about the studied phenomenon. The detailed answers to your new questions and comments are shown in the following Annex.
Kind regards.
The authors

Reviewer 3 Report
Comments and Suggestions for Authors
The manuscript "Measuring DNI with a new radiometer based on Optical Fiber and Photodiode" by Carballar et. al. proposed a new equipment for measuring solar irradiance. The manuscript described the design, working mechanism, and the experimental verification of it. Overall, the experiment supports the design, and the presentation is clear. Meanwhile, I would like to see the clarification on several concerning aspects before publishing.
1. The author admits the importance of the correction factor, and the factor in this manuscript is obtained by calibration from a mature equipment. In real use cases, it is very possible that the calibration is done once, and then used for a long time thereafter. In these situations, how robust is the CF against changing environment (like temperature and weather)? For example, the author mentioned CF may change between mid-day and sunrise/sunset. Would it also change in sunny/rainy/cloudy days (as the spectra are be different)?
2. I understand the author used a very long optical fiber to isolate the light collection and detection. In all the equations, the effect of such long fiber is captured by 2 parameters, alpha and L. I wonder how complete this description is. For example, would alpha change with respect to temperature? Will light leak at kinks of the fiber?
3. I would like the author to explain a bit more on equation (3). I can understand the integral part. But why we need a R(lambda_i) in the denominator, and how is this wavelength chosen?
4. The title "DNI" indicates the "normal" solar power. For such a thin optical fiber, how would this "normal" be guaranteed? How sensitive it is when considering installation error, like a fraction of angle off from ideal direction?
Overall, I think the method proposed in the manuscript lacks some practical consideration. I would like to see the authors emphasize on the above points before recommending to publish it.
Comments on the Quality of English LanguageThe quality of English is acceptable. Please do a more detailed proof reading before submission. Several modifications I would suggest include:
Line 94, "indices" may be a more proper word comparing with "indexes"
Eq (1), sen may should be sin.
Author Response
Dear Reviewer #3,
We appreciate your comments and revisions that helped us to improve the manuscript and our knowledge about the studied phenomenon. The detailed answers to your new questions and comments are shown in the following Annex.
Kind regards.
The authors

Round 2
Reviewer 2 Report
Comments and Suggestions for Authors
Dear authors,
Thank you so much for your answer and changes in the new version of the paper but, a correct error treatment is still missing on the paper:
What is the instrumental error of the pyrheliometer?
What is the instrumental error of the used photodiodes and optical power meter?
With this instrumentation (photodiodes and optical power meter) you are measuring luminosity basically. How the combined error of both devices is translated to the calculated DNI?
Please include this calculation on the paper. At the end, you must calculate a final uncertainty of the calculated DNI measured with the proposed device (in percentage). This percentage will ensure the quality of the proposed device.
Author Response
Dear Reviewer #2,
We appreciate your comments and suggestions that helped us to improve the manuscript and our knowledge about the proposed instrument. The detailed answers to your recommendations are shown in the following Annex.
Kind regards.
The authors

Reviewer 3 Report
Comments and Suggestions for Authors
My questions have been answered and I'm satisfied with most of the answers. For my last question, I still think the author should comment on the possible alignment issue or mechanical instability problem, as the title explicitly indicates "normal".
Other than that, I don't have further comments and the manuscript can be published with the minor modification.
Author Response
Dear Reviewer #3,
We appreciate your comments and suggestions that helped us to improve the manuscript and our knowledge about the proposed instrument. The detailed answers to your questions and comments are shown in the following Annex.
Kind regards.
The authors
